# CYP1-Activation and Anticancer Properties of Synthetic Methoxylated Resveratrol Analogues

**DOI:** 10.3390/molecules29020423

**Published:** 2024-01-15

**Authors:** Ketan C. Ruparelia, Keti Zeka, Kenneth J. M. Beresford, Nicola E. Wilsher, Gerry A. Potter, Vasilis P. Androutsopoulos, Federico Brucoli, Randolph R. J. Arroo

**Affiliations:** 1Leicester School of Pharmacy, De Montfort University, The Gateway, Leicester LE1 9BH, UK; kruparel@dmu.ac.uk (K.C.R.); kjmberesford@gmail.com (K.J.M.B.); nicola.wilsher@astx.com (N.E.W.); rrjarroo@dmu.ac.uk (R.R.J.A.); 2Zayed Centre for Research into Rare Disease in Children, University College London, London WC1E 6BT, UK

**Keywords:** stilbenes, CYP1A1, CYP1B1, prodrug, Wittig reaction, breast cancer

## Abstract

Naturally occurring stilbenoids, such as the (*E*)-stilbenoid resveratrol and the (*Z*)-stilbenoid combretastatin A4, have been considered as promising lead compounds for the development of anticancer drugs. The antitumour properties of stilbenoids are known to be modulated by cytochrome P450 enzymes CYP1A1 and CYP1B1, which contribute to extrahepatic phase I xenobiotic and drug metabolism. Thirty-four methyl ether analogues of resveratrol were synthesised, and their anticancer properties were assessed, using the MTT cell proliferation assay on a panel of human breast cell lines. Breast tumour cell lines that express CYP1 were significantly more strongly affected by the resveratrol analogues than the cell lines that did not have CYP1 activity. Metabolism studies using isolated CYP1 enzymes provided further evidence that (*E*)-stilbenoids can be substrates for these enzymes. Structures of metabolic products were confirmed by comparison with synthetic standards and LC-MS co-elution studies. The most promising stilbenoid was (*E*)-4,3′,4′,5′-tetramethoxystilbene (DMU212). The compound itself showed low to moderate cytotoxicity, but upon CYP1-catalysed dealkylation, some highly cytotoxic metabolites were formed. Thus, DMU212 selectively affects proliferation of cells that express CYP1 enzymes.

## 1. Introduction

Cancer, a generic name for over 200 different diseases, is defined as the uncontrolled proliferation and spread of cells in which the physiological factors governing normal cell growth have become ineffective [1]. Natural products from plants have been the most consistently successful source of drug leads for anticancer drugs [2,3]. Over the last two decades, compounds with a stilbene nucleus have attracted considerable interest [4]. Semisynthetic phosphorylated derivatives of the natural compound combretastatin A4, a polymethoxy (*Z*)-stilbenoid, originally extracted from the African bush willow tree (*Combretum caffrum*), have gone through phase I and phase II clinical trials [5]. These pro-drugs are cleaved into the natural form by endogenous phosphatases and then taken up by cells. The lead compound, CA4P (fosbretabulin), showed potential to stabilise a tumour in a phase II clinical trial (ClinicalTrial.gov Identifier: NCT00077103). Ombrabulin (AVE8062), which was marketed by Sanofi Aventis, is an analogue of CA-4 with (S)-amino-hydroxypropanamide [-(NH-CO-NH_2_OH)] modification in the B-ring (Figure 1). However, its development for use in ovarian cancer was discontinued in 2013 after disappointing trial results [6].

Resveratrol, (*E*)-4,3′,5′-trihydroxy stilbene, commonly found in its glycoside form (polydatin or piceid) in red wine, grape skins, peanuts and cranberries, has also attracted considerable interest as a potential anticancer drug [7,8,9,10,11,12]. Resveratrol is classified as a phytoestrogen because of its structural similarity to endogenous oestradiol.

In fact, a wide range of naturally occurring polyphenols, notably flavonoids, are labelled as phytoestrogens since they interact with receptors and enzymes that would normally have oestradiol as their natural substrate. Several reports have confirmed that flavones and flavonols interact with CYP1 enzymes, either as substrates, inhibitors, or inducers [13,14,15,16].

The cytochromes P450 (CYPs) are a super family of enzymes, typically involved in the bioactivation and detoxification of xenobiotics and endogenous substrates. Metabolising enzymes are conventionally classified as phase I and phase II enzymes. CYPs are responsible for the majority of phase I drug metabolism [17,18]. Most cytochrome P450 enzymes are active specifically in the liver, but CYP1A1 is expressed extrahepatically, e.g., in the brain, endocrine tissues, lungs, gastrointestinal tract, urinary tract, and in tissues of the male and female reproductive tract. CYP1B1 is thought to play a role in early embryonic development, but in adults is a marker for tumour development [19,20].

Binding studies indicated that the B-ring of flavones is placed near the prosthetic group of CYP1 enzymes, with C-3′ and C-4′ being closest to the reactive heme iron [14,16]. The A-ring of flavones is positioned further away from the prosthetic group, and is thus less prone to regioselective conversions. It may play a role in stabilising flavones as substrates in the enzyme pocket. Flavones that contain multiple methoxy groups tend to have a higher metabolic turnover by CYP1 enzymes [21,22]. Studies on flavones as potential substrates for CYP1 enzymes have indicated the 4′-OMe substitution on the B ring as a target for regiospecific *O*-dealkylation [16,21,22,23,24,25]. Further, if a 4′-OH substitution is present on the B ring, CYP1 enzymes catalyse regiospecific hydroxylation of the C-3′ [23,26]. In all cases, the result of the CYP1-mediated bioconversion is the formation of a catechol-type substitution on the B-ring with a concurrent increase in cytotoxicity. This structure–activity relationship reflects a well-known pattern: tyrphostin A1, which bears a methoxy group in the *para*-position of its aromatic ring (Figure 2), is completely inactive as a tyrosine kinase inhibitor, whereas increasing the number of hydroxyl substituents on the ring resulted in a 35-fold increase in potency [27].

The structure of resveratrol can be mapped onto that of oestradiol (Figure 3) with the C-3 in oestradiol corresponding to the C-4′-position in resveratrol. Previous studies in our lab have shown that CYP1B1, an enzyme that has oestradiol as its natural substrate, also catalyses the conversion of resveratrol into piceatannol (*E*)-3, 4,3′,5′-tetrahydroxy stilbene [28]. Piceatannol differs from resveratrol by having one additional aromatic hydroxyl group, thus creating the catechol motif that is commonly found in kinase inhibitors [29,30].

The pharmacokinetic profile of resveratrol is rather poor. This natural product is rapidly and extensively metabolised to its major phase II metabolites, including resveratrol sulphates and glucuronides, which are subsequently excreted from the body [31].

Here, we report a series of synthetic analogues of resveratrol that were designed as potential pro-drugs to be selectively activated by CYP1 enzymes. Activation of the drugs via dealkylation is expected to result in the formation of compounds possessing motifs that are characteristic of tyrosine kinase inhibitors or anti-mitotic activity. Various methoxy substitution patterns were compared to assess their function in determining the efficacy of enzyme-catalysed conversions. Particular attention was paid to the 3′,4′,5′-trimethoxy substitution pattern since this motif features in many tubulin-binding agents such as colchicines and combretastatin A4. The polymethoxy *cis*- and *trans*-stilbenes were designed to mitigate resveratrol phase II metabolisation events and undergo selective dealkylations to form active metabolites, which are expected to act as tyrosine kinase inhibitors or antimitotic agents. The new *cis*-/*trans*-stilbene derivatives were investigated for antiproliferative activity using a panel of human breast cell lines. The panel consisted of MCF7, an oestrogen receptor-positive cell line (e.g., ER-positive tumour cell line with the ability to process oestradiol, via oestrogen receptors in the cell cytoplasm, but with very low levels of CYP1 expression). MCF7 cells can be induced with the aromatic hydrocarbon receptor (AhR) agonist 2,3,7,8-tetrachlorodibenzo-*p*-dioxin (TCDD), after which they express high levels of CYP1A1 with a concomitant increase in oestrogen metabolism [32,33,34]. The cell panel also included the MDA-MB-468 (MDA468) cell line, which is an ER-negative tumour cell line that constitutively expresses both CYP1B1 and CYP1A1 [32]. Finally, the control cell line was MCF10A, a non-tumour ‘normal’ cell line which has no basal CYP expression [34,35].

Ideally, compounds tested should show little cytotoxicity in control MCF10A cells, whereas in MDA468 cells, the compounds after activation by CYP1 enzymes are expected to become highly cytotoxic. The difference in cytotoxicity is referred to as tumour selectivity (TS). Further, little activity should be expected in MCF7 cells until after the induction of CYP1 activity by TCDD. In this case, the difference in activity between uninduced and TCDD-induced cells is referred to as the activation factor (AF).

## 2. Results

### 2.1. Synthesis of New Resveratrol Analogues

A library of *trans* and *cis* stilbene geometric isomers was synthesised using a standard Wittig reaction procedure (Figure 1). The benzyltriphenyl phosphonium chloride substrates were obtained either as commercially available substituted benzyl chlorides, or as benzyl alcohols, which were converted to their corresponding alkyl chloride using triphenylphosphine and carbon tetrachloride via the Appel reaction mechanism. Flash column chromatography on silica gel afforded the *cis*- and *trans*-stilbenes. Individual compounds were characterised using NMR and LC-MS analysis.

### 2.2. Determination of Antiproliferative Activity

The antiproliferative activity of the resveratrol analogues was investigated using a human breast cell line panel, which has been characterised for CYP1 expression in our laboratories. IC_50_ values for both *cis*- and *trans*-isomers were determined using a standard 3-[4,5-dimethylthiazol-2-yl]-2,5-diphenyl tetrazolium bromide (MTT) cell viability assay [36] and are shown in Table 1 (*trans*-stilbene analogues) and Table 2 (*cis*-stilbene analogues).

Little or no bioactivity was shown by any stilbenoid analogues with either a single -Me or -OMe, or *meta*- or *para*- double -OMe substitutions in the B-ring. A 3,4,5-trimethoxy substitution on the A-ring causes a significant increase in cytotoxicity. In the case of the *trans* stilbenoid DMU212, the cytotoxicity was particularly increased in cell lines expressing either CYP1B1 or CYP1A1 (i.e., MCF7 after TCDD induction, and MDA-468). The *trans* stilbenoid DMU212 was over 4000 times more toxic to the tumour cell line MDA-468 (IC_50_ = 10 nM) than to the non-tumourigenic cell line MCF10A (IC_50_ = 4.3 µM), with a remarkable tumour selectivity (TS) value of 4300. Interestingly, DMU213, which is a *cis* isomer of DMU212, was 500-fold less toxic to MDA-468 compared to DMU212 and showed similar cytotoxicity to both MDA-468 and MCF10A with a TS value of only four.

The cytotoxicity was significantly reduced in the 2,3,4-trimethoxy-substituted *trans* (DMU547) and *cis* (DMU548) stilbenoids, with IC_50_ MDA468 values of 4.5 and 30 µM, respectively.

Substitution of the 4′-OMe on the A-ring by a 4′-OEt (DMU260 and DMU261) resulted in a marked increase in cytotoxicity, but there was little difference in activity against tumour cell lines MCF7 or MDA468 and the non-tumour cell line MCF10A.

Encouraged by cytotoxicity results obtained with DMU212, we sought to synthesise and test its dihydrostilbene analogue, DMU224, which was prepared and tested for cytotoxic activity. DMU224 was prepared in moderate to high yield by catalytic hydrogenation of stilbene DMU212 to reduce the ethene bond present between the two aryl rings to an ethylene bond (Figure 2).

DMU224 showed to be highly toxic for all cell lines including non-tumour MCF10A cells, and the IC_50_ values were 0.06 µM for both MCF7 and TCDD-induced MCF7, 0.005 µM for MDA-468, and 0.06 µM for MCF10A.

In order to investigate the importance of the 4-methoxy substituent in the A-ring of DMU212, corresponding unsubstituted (DMU507) and 3,4-methylenedioxy (DMU220) A-ring stilbenes were prepared and assayed. In comparison to DMU212, DMU507 showed significantly increased IC_50_ values for all cell lines (i.e., IC_50_ > 10 μM). In comparison to DMU507, its respective *cis* isomer DMU508 was relatively more toxic to all cell lines tested (i.e., IC_50_ ranging from 4 to 9 μM).

Compared to DMU212, its 3,4-methylenedioxy analogue DMU220 showed slightly increased IC_50_ values for all cell lines. In the MDA468 assay, DMU220 had given a TS value of four, and IC_50_ values of 0.99 µM and 3.9 µM for the MDA468 and MCF10A cells, respectively. The corresponding *cis* isomer of DMU220, DMU299, was relatively more toxic to both induced and non-induced MCF7 cells with an AF of one and an IC_50_ value of 0.05 µM in both cell lines. In the MDA468 assay, this compound was relatively less toxic than DMU220 giving a TS value of one and an IC_50_ value of 20 µM. The low AF and TS values were surprising as these methylenedioxy analogues were predicted to undergo CYP1-catalysed metabolism to form toxic catechol metabolites via dealkylation of the methylenedioxy group.

A naphthyl A-ring system was introduced to investigate the effect of additional aromatic rings on cytotoxicity compared to the mono-aromatic A-ring in DMU212. It can be seen that replacing the 4-methoxyphenyl unit (A-ring) of DMU212 with either 6-methoxynaphthyl (DMU567) or 4-methoxynaphthyl (DMU569) moieties led to analogues with significantly increased IC_50_ values for all cancer cell lines giving TS values of three and nine, respectively. DMU567 was toxic to MCF10A cells, whilst DMU569 was essentially non-toxic to the same cell line (IC_50_ = 100 µM) resulting in decreased activity in the MDA468 cells compared to DMU212. As can be noted, modification of the stilbenoids’ A-ring led to an increase in the IC_50 MDA468_ value of DMU212 from 0.001 µM (DMU212) to 0.04 µM (DMU567) and 4.5 µM (DMU569), respectively. DMU568, in comparison to its corresponding *trans* isomer DMU567, was more toxic to both induced and non-induced MCF7 cells. DMU568 showed similar potency in MDA468 cells in comparison to DMU213, which is the *cis* isomer of DMU212, but it was relatively non-toxic to MCF10A cells, giving a TS value of 100.

Further, dimethoxy-substituted A-ring stilbenes were prepared, including the respective *cis* isomers, to investigate the importance of a 4-methoxy substituent in the A-ring of DMU212. In comparison to DMU212, the *trans* DMU2417 isomer bearing a 3,4-dimethoxy substitution on the A-ring showed significantly increased IC_50_ values for all cell lines (i.e., IC_50_ > 9 μM). Similar potencies were seen with its corresponding *cis* isomer DMU1024. However, *cis* isomer DMU222 with a 2,4-dimethoxy substitution on the A-ring exhibited high cytotoxicity in all cell lines, albeit it was 10-fold more active in MDA468- and MCF7-induced cells (0.005 μM) compared to MCF10A and MCF7 cells (0.06 μM), suggesting that this compound is a poor candidate for further investigation.

Changing the 4-methoxy substitution in the A-ring of DMU212 with the 4-ethoxy group, as in DMU260, gave a low AF value of one, and IC_50_ values of 5 µM for both induced and non-induced MCF7 cell lines. In the MDA468 assay, DMU260 had a TS value of two and an IC_50_ value of 1.8 µM. In comparison, in the same assay, DMU212 gave an AF of 22 and TS of 4300. DMU261, the corresponding *cis* isomer of DMU260, exhibited potent toxicity to all cell lines tested including the MCF10A cells. The low AF and TS values observed for the ethoxy substituted stilbenes were surprising, as these ethoxy A-ring stilbenes were anticipated to undergo an initial CYP1-mediated activation. This would result in the dealkylation of the ethoxy group to form a 4-hydroxy primary metabolite and with the subsequent formation of a secondary catechol metabolite following 3-hydroxylation by CYP1 enzymes.

The 4-halogenated A-ring stilbenes were synthesised as analogues of DMU212 and assayed for their anticancer activity. They were synthesised to study the effect of switching the CYP1 metabolism to the B-ring where demethylation is possible. In comparison to DMU212, the chloro-substituted DMU555, the bromo-substituted DMU557 and the 4-iodo-substituted DMU509 all showed significantly increased IC_50_ values for all cell lines tested. The corresponding *cis* isomers DMU556, DMU558 and DMU510 were relatively more toxic to all cell lines tested. DMU501 with a 4-iodo substitution was found to be the most potent of the series with, an IC_50_ of 0.06 µM for both induced and non-induced MCF 7 cells, and an IC_50_ of 0.006 µM for MDA468 cells is seen.

### 2.3. CYP1-Catalysed Bioconversion of DMU212

The most promising of the synthetic resveratrol analogues was DMU212, which showed only moderate toxicity towards control cells (IC_50_ = 4.3 µM), and a 4300-fold tumour selectivity. DMU212 was designed to be selectively bioactivated by CYP1B1, which is a cytochrome P450 enzyme reportedly expressed in a variety of hormone-related tumours. Indeed, the IC_50_ values for DMU212 were significantly lower in cell lines expressing CYP1 enzymes, i.e., MDA468 which constitutively expresses CYP1 enzymes and MCF7 which express CYP1 only after TCDD-induction, compared to cells not expressing CYP1, i.e., cell lines MCF-10A and MCF-7 without induction (see Table 1). The effect of DMU212 on MDA468 and cells was abated by the presence of known CYP1 inhibitors acacetin and α-naphthoflavone (Figure 4). This is further evidence that the toxicity of this compound in the cancer cell lines is due to CYP1-mediated activation.

DMU212 was shown to be a substrate for CYP1 enzymes. Bioconversions were preferentially catalysed by CYP1A1 and CYP1A2, and to a lesser extent by CYP1B1. The bioconversion profile of DMU212 showed the formation of three primary metabolites confirmed by co-elution studies with synthesised authentic standards as DMU214, resulting from 3′-hydroxylation and DMU 291 and DMU281, resulting from para-O-demethylation (Figure 5).

The *cis*-isomer of DMU214, DMU215 (combretastatin A4), was not detected in incubates of DMU212 with CYP1B1. This result indicates that *cis-trans* isomerisation does not occur during enzymatic oxidation of DMU212.

Further metabolism by CYP1 enzymes showed the formation of secondary metabolites identified as DMU295, DMU283 and DMU293 from co-elution studies (Figure 6). DMU 212 was preferentially metabolised by CYP1A1 rather than by CYP1B1 to the active catechol metabolite DMU214 a predicted tyrosine kinase inhibitor and two *para O*-demethylated metabolites DMU291 and DMU281.

If our hypothesis behind the rationale for prodrug design and CYP1-mediated bioactivation is correct, then the identified metabolites should show toxicity to all the cell lines when tested for their in vitro cytotoxicity as they are already bioactivated to toxic species. Only DMU214 and DMU283 were found to be highly toxic to all cell lines tested (Table 3).

## 3. Discussion

A series of synthetic and naturally occurring *cis*- and *trans*-stilbenoids was screened for potential anticancer drug activity. Cytotoxicity results indicate that *trans*-stilbenoid DMU212 (3,4,5,4′-tetramethoxystilbene) may be a potential tumour-selective anticancer agent with therapeutic potential for the treatment of CYP1 enzyme-expressing tumours (i.e., oestrogen-dependent tumours). The differential CYP1A1 and CYP1B1 expressions in normal tissues and tumour tissues may provide an opportunity for the development of tumour-selective drugs for the effective treatment of a variety of different cancers [32,37]. DMU212 was shown to have little toxicity towards cells that did not express CYP1 activity. However, in CYP1-expressing cells, the stilbenoid would act as an enzyme substrate that is converted into cytotoxic metabolites, notably DMU 214 (3,4,5,4′-tetramethoxy-3′-hydroxy stilbene) and DMU283 (3,4,5-trimethoxy-3′,4′-dihydroxy stilbene).

The *cis*-isomer of DMU212, DMU213, also showed appreciable toxicity towards CYP1-expressing MDA468 cells (IC_50_ 0.5 µM) and TCDD-induced MCF7 cells (IC_50_ 0.44 µM). However, this *cis*-stilbenoid was more toxic to MCF10A cells (IC_50_ 2 µM) than its *trans* counterpart (IC_50_ 4.3 µM). The high potency of DMU213 is not that surprising since this *cis*-stilbenoid may be regarded as a prodrug of the known cytotoxic agent combretastatin A4 [38,39].

The dihydrostilbenoid analogue of DMU212 and DMU213, DMU224, showed sub-micromolar cytotoxicity against MDA468 cells and TCDD-induced MCF7 cells, but was highly toxic to all cell types, including the non-tumour cell line MCF10A (IC_50_ = 0.09 µM). The MTT assay results indicate that the presence of the double bond in DMU 212 is essential for optimal selectivity against CYP1 expressing tumour cells. Interestingly, two or more planar rings and the trimethoxy moiety are reportedly important in tubulin-binding agents such as colchicine, combretastatins and podophyllotoxin [38].

DMU212 was also evaluated for cytotoxic activity in the aggressive, triple negative breast tumour cell line MDA231. This cell line has been reported to only weakly express CYP1A1 but constitutively express CYP1B1 [33]. DMU212 showed no toxicity towards the MDA231 cell line (IC_50_ > 100 µM). The non-toxicity towards MDA231 cells but high potency towards MDA468 cells indicates that the mechanism of activation of DMU 212 may be via CYP1A1 rather than CYP1B1. However further studies would be required to confirm this statement.

Of the CYP1 conversion products of DMU212, only DMU214 and DMU283 were found to be highly toxic to all the cell lines tested (sub-micromolar IC_50_ values). This may be due to the increased lipophilicity of DMU214 and DMU283 which maintain the trimethoxy moiety, putatively improving transport across the cell membrane.

Although DMU212 demonstrated selectivity for tumour cells in our in vitro assays, clinical application of this prodrug may be hampered since the compound is also a substrate for CYP1A2 which is predominantly expressed in the liver. However, considering that the resulting kinase inhibitors seem to block the cell cycle rather than cause apoptosis [21,24,40,41], and that cells in the adult liver only divide rarely, damage to the liver may be limited. These issues may be addressed in appropriate in vivo preclinical trials. Early results of mouse experiments have indicated that resveratrol is more readily absorbed than DMU212 but is then rapidly converted into sulphate or glucuronate conjugates and removed from the body. In contrast, DMU212 underwent metabolic hydroxylation or single and double O-demethylation, but the metabolites remained in systemic circulation for a bit longer [42]. Further mouse experiments showed that DMU212, when given as 0.2% (*w*/*w*) of the diet for a period of 14 weeks, resulted in a steady state of up to 0.7 nmol/g tissue in the liver [43], without any obvious signs of hepatoxicity.

## 4. Materials and Methods

### 4.1. Materials

All solvents and chemicals were used as purchased without further purification. The ^1^H and ^13^C NMR spectra were recorded on a Bruker Avance AV400 NMR spectrometer at 30 °C. Chemical shifts were reported in d units (ppm) relative to either TMS or the residual solvent signal. IR spectra were recorded as KBr discs on a Perkin-Elmer 298 spectrophotometer. HRMS was performed using a Thermo Scientific LTQ Orbitrap XL at the EPSRC National Mass Spectrometry Service Centre (Swansea, UK). Melting points (uncorrected) were determined using a Gallenkamp melting point apparatus in open glass capillary tubes. TLC was performed on Merck Silica Gel 60F_254_-coated plates. Plates were visualised under UV light (254/366 nm) and stained with either 2,4-dinitrophenylhydrazine, iodine or phosphomolybdic acid. Fluka Silica gel 60 (30–45 m) was used for flash chromatography. The purities of compounds were determined using either elemental analyses (C, H, N) or HPLC. The former was performed on a CE440 elemental analyser by Warwick Analytical Services. Results were within ±0.4% of the theoretical values. HPLC was carried out on a Perkin Elmer 200 series chromatography system using a HAISIL 100C18 (250 × 4.6 mm) column eluting with 60% acetonitrile/water (Flow rate: 1.0 mL/min). All samples had purity greater than 95%.

Cell lines MCF7, MDA231, MDA468 and MFC10A were purchased from the American Type Culture Collection (Manassas, VA, USA).

### 4.2. General Synthetic Procedures

The synthesis of stilbenes by Wittig reaction, the most widely used approach, typically involves the condensation of equimolar quantities of substituted benzaldehyde with phosphorus ylide generated in situ when a phosphonium salt is treated with a strong base such as n-butyllithium Figure 1 (See Appendix A for details).

Appropriately substituted benzyltriphenyl phosphonium chlorides (1 equivalent) in anhydrous tetrahydrofuran (THF) were cooled to −20 °C under an atmosphere of nitrogen and n-butyllithium in hexane solution (1 equivalent) was then added and stirred for 20 min; this resulted in a colour change from colourless to a deep red, marking the formation of the ylide. Then the appropriately substituted benzaldehyde (1 equivalent) was added resulting in a colour change from red to pale yellow and the mixture was stirred for 24 h. The reaction was monitored by TLC (*cis*-stilbene, and *trans*-stilbene appeared as a dark spot and bright blue spot, respectively, and the *cis*-isomer always had a higher TLC Rf value than the *trans*-stilbene, when visualised under UV light). Stilbenes were formed as a mixture of *cis-trans* geometric isomers with triphenylphosphine oxide as a by-product. Purification by multiple flash column chromatography was necessary to separate stilbene isomers and to separate the product triphenylphosphine oxide from the mixture formed during the Wittig reaction. This purification and isolation step was time consuming but allowed individual isomers to be isolated with high purity, as confirmed by TLC. Enough was isolated (at least 50 mg) to allow full chemical characterisation and assessment of their anticancer activity.

Where a phosphonium salt is not available, it can be readily prepared by refluxing the appropriate benzyl halide and triphenylphosphine in toluene for 2–4 h, as employed by [44]. The resulting salts can be collected by filtration and used without further purification. If the desired benzyl halide was unavailable, triphenylphosphine and carbon tetrachloride mixture were successfully used to convert benzyl alcohols to corresponding benzyl halides. Classic Wittig reactions to stilbenes gave both *cis*- and *trans*-stilbenes with the *trans*-isomers generally being obtained as solid powder or crystalline solid and the *cis*-stilbenes as viscous oil. Generally, almost equal amounts of both the *cis*- and *trans*-stilbenes were obtained. The purity of all the stilbenes prepared was established by TLC and/or CHN microanalysis. The structures of all the compounds were confirmed by ^1^H-NMR and ^13^C-NMR spectrometry. The *cis* and the *trans* geometries of the stilbenes were assigned by the characteristic ^1^H-NMR coupling constants of the olefinic protons that appear as a set of two doublets with about 12 Hz for *cis*-isomers and 16 Hz for *trans*-isomers (Lit: J_cis_ = 5–14 Hz; J_trans_ = 11–19 Hz). The *cis*-stilbene protons generally gave signal up field at lower chemical shifts than the corresponding *trans*-stilbene protons. The mass spectra (MS) with corresponding [M^+^] or [M + H]^+^ peaks for each stilbene and accurate mass analysis confirmed the expected molecular formula.

The aryl methyl halides, such as benzyl chloride, are heated with trialkyl phosphite such as triethylphophite to afford the phosphonate ester via an Arbusov reaction. The reaction proceeds by way of nucleophilic attack by the lone pair on the phosphate on the alkyl group of the alkyl halide via phosphonium intermediate which rearranges to form phosphonate ester and alkyl halide. Horner–Wittig reaction of aryl aldehyde with phosphonate anion, an intermediate which is formed in situ by the reaction of phosphonate esters with strong bases such as sodium hydride or sodium *tert*-butoxide, generates the *trans* exclusive stilbene (See Appendix A for details).

The *trans* stilbene stereochemistry is determined by the rapid kinetics of the intermediate oxaphosphotane ring closure, where the *cis* pathway is much slower than the *trans*. The generated library possesses either varied methoxy and/or methylenedioxy groups, allowing either direct aromatic hydroxylation or O-demethylation or demethylenation to form catechol metabolites. The presence of lipophilic groups such as methoxy and methylenedioxy would serve to avert the rapid degradation associated with the phenolic hydroxyl groups and hence improve bioavailability.

The proposed CYP1-catalysed metabolism of the methylenedioxy substituent formed a catechol metabolite, as predicted by mapping onto oestradiol, where O-demethylation of the methylenedioxy motif initiates spontaneous rearrangement to generate formate ester and on hydrolysis with a loss of the formate group produce the catechol metabolite. CYP1-catalysed metabolism by demethylation is via formation of an unstable hemi-acetal which upon hydrolysis with a loss of the formaldehyde, produces the demethylated metabolite, similar to what has been reported for the oestradiol catechol [45]

### 4.3. General Procedure for the Preparation of Stilbenes

To a stirred suspension of the appropriate benzyltriphenyl phosphonium chloride (2.4 mmol) in anhydrous THF (20 mL), at −20 °C under nitrogen, a solution of n-butyllithium in hexanes was added dropwise (1.49 mL, 2.4 mmol, 1.6 M in hexane). The resulting red suspension was stirred for 20 min at −20 °C and then the appropriate benzaldehyde (2.4 mmol), in anhydrous THF (10 mL), was added dropwise. The reaction was stirred for 1 h at −20 °C and then allowed to warm to room temperature and stirred overnight. The reaction mixture was quenched with ice water (40 mL) and extracted with ethyl acetate (3 × 20 mL). The combined organic extracts were washed with water (2 × 20 mL) and brine (2 × 20 mL) and dried over anhydrous magnesium sulphate. The solvent was removed in vacuo to afford a mixture of *cis*/*trans* isomers. Flash column chromatography (petroleum ether 40–60 with an increasing gradient of ethyl acetate (20–40%, silica gel 60, 20–45 µm) afforded the *cis*- and *trans*-stilbenes, respectively.

The dihydrostilbenoid DMU224 was prepared by catalytic hydrogenation of stilbene DMU212, to transform the ethene bond to ethylene bond present between the two aryl rings in the presence of 10% palladium on carbon in anhydrous THF. The mixture was stirred at room temperature for 20 min, monitored by TLC to show the appearance of a new non-polar spot in comparison to DMU212. Removal of catalyst by filtration followed by trituration afforded a white solid in 48% yield.

### 4.4. Biological Assays

MDA468 cells represent an aggressive multi-drug-resistant tumour type and are ER negative but are aryl hydrocarbon (Ah) receptor positive in the presence of TCDD [46]. MDA468 show high CYP1 constitutive activity; Western blotting experiments indicated that CYP1A1 was the main active cytochrome P450, with only a trace signal for CYP1B1 [21].

MCF7 cells are both ER positive and Ah receptor positive and have been well documented for their CYP1 expression, and it is known that treatment with TCDD elevates CYP1A1 and CYP1B1 expression [35]. The cells were treated with DMSO (0.1%) for 24 h for constitutive expression, whereas for inducible CYP1 expression, the cells were treated with TCDD (10 nM) for 24 h.

MDA231 breast cancer cell line represents an aggressive tumour type. Treatment of this ER-negative and Ah receptor-positive cell line with TCDD has been reported to only weakly induce CYP1A1 mRNA expression but enhance the constitutive CYP1B1 expression [33]. Other researchers have reported the ratio of active CYP1A1/CYP1B1 protein as 16/84 ratio in MDA231 [32].

MCF10A cells are non-tumourigenic human mammary epithelial in origin which have been spontaneously immortalised without chemical intervention. CYP1 activity in MCF10A cells is negligible [21]. Significant toxicity to these cells could suggest non-selectivity to tumour cells over healthy cells. Thus, MCF10A cell line is commonly used as a control in cytotoxicity assays.

#### 4.4.1. MTT Cytotoxicity Assay

A standard MTT (3-[4,5-dimethylthiazol-2-yl]-2,5-diphenyl tetrazolium bromide) cell viability assay [47] was employed to evaluate the cytotoxicity of the stilbenes prepared. The MTT assay is a rapid quantitative spectrophotometric method for determining cell growth (viability) due to necrosis or apoptosis in monolayer-cultured cell lines in response to external factors. It is widely employed to evaluate the cytotoxicity of potential medicinal agents [36].

MCF7 cells were grown in RPMI-1640 medium with phenol red and 10% (*v*/*v*) heat-inactivated (heated to 56 °C for 45 min) foetal calf serum. MDA-468 cells were grown in RPMI-1640 medium with 10% (*v*/*v*) heat-inactivated foetal calf serum and L-glutamine (2 mM) without phenol red. MCF-10A cells were grown in Dulbecco’s modified Eagle’s medium/ Ham’s F-12 medium (1:1) with 5% (*v*/*v*) heat-inactivated foetal calf serum, epidermal growth factor (20 ng/mL), insulin (10 mg/mL) and hydrocortisone (500 ng/mL). Cells were maintained at 37 °C, 5% CO_2_/95% air with 100% humidity and passaged every 2–3 days using trypsin EDTA solution (0.25% *w*/*v*). Adhered cells at sub confluence were harvested for experimental use.

Aliquots of cell suspensions (100 µL at 2 × 10^3^ cells per ml) were dispensed into sterile 96-well microtiter flat-bottomed plates. For the MDA468 and MCF10A cell lines, the plates were incubated at 37 °C, 5% CO_2_/95% air with 100% humidity for 24 h prior to the addition of test compounds. For MCF7 cells, after allowing approximately 4 h for cells to adhere, 100 µL of medium containing TCDD (from 100 mM stock in DMSO) or medium with 0.2% (*v*/*v*) DMSO was added to give a final concentration of 10 nM TCDD and 0.1% (*v*/*v*) DMSO. After 24 h, the medium was aspirated and replaced with fresh medium (100 µL) before the addition of test compounds.

Then, 100 µL aliquots of test compounds were added within 30 min into the wells from a 100 mM stock solution in DMSO and serially diluted to give final concentrations of 100, 30, 10, 3, 1, 0.1, 0.93, 0.01, 0.003, 0.001 and 0.0003 µM. The final concentration of DMSO did not exceed 0.1% *v*/*v* in each well. The cells were allowed to grow for 96 h at 5% CO_2_, 37 °C, to give 80–90% confluence in the control wells, after which 50 µL of 2 mg/mL MTT (final concentration 0.4 mg/mL) in sterile phosphate buffer was added to each well and the plates were further incubated for 2 h. All medium was aspirated and the formazan precipitate generated by viable cells was solubilised by 150 µL of DMSO. All the plates were vortexed and the absorbance at 540 nm was determined using a SpectraMax M5 plate reader (Molecular Devices, Wokingham, UK).

Results were expressed as a percentage of the control value versus the negative logarithm of the molar drug concentration range using GraphPad Prism (Version 5.00). Relative toxicities of each compound within each cell line were expressed as 50% of growth inhibition (IC_50_). All determinations were carried out in quadruplicate.

#### 4.4.2. Enzyme Assays

The CYPs isozyme panel used in this study were prepared from transformed insect cells expressing human CYPs (CYP1A1, CYP1A2 or CYP1B1) with co-expression of human NADPH reductase (Supersomes^TM^). Control microsomes (inactive microsomes) were prepared from insect cells but without the human CYP cDNA. The general procedure was as follows: DMU212 was separately incubated with CYP1A1, CYP1A2, CYP1B1, CYP3A4 and non-CYP-expressing Supersomes^TM^ in the presence of phosphate buffer and NADPH at 37 °C. Samples were taken at regular intervals of 5 min over a 20 min period, and the enzymatic reaction in the sample was terminated immediately. Following centrifugation, the supernatants were removed and analysed by HPLC.

#### 4.4.3. HPLC Analysis

Separation of DMU212 and its potential metabolite standards was achieved using the following analytical conditions: Phenomenex Ultracarb ODS (30) analytical column, 250 × 4.6 mm at 40 °C; mobile phase 45% CH_3_CN, 10% propan-2-ol, 45% 50 mM ammonium acetate; flow rate 1 mL/min (and fluorescence detection with an excitation at λ = 335 nm and emission at λ = 395 nm). After 10 min of analysis, a linear gradient was initiated with the acetonitrile content rising to 70% over 5 min and with the 50 mM ammonium acetate decreasing accordingly. An equilibration time of 6 min was allowed between each sample. Analysis was conducted with a Jasco (Jasco UK limited) analytical system consisting of an AS841 autosampler, PU980 pump and LG low-pressure tertiary mixing valve, FP1500 fluorescence detector (JASCO UK Ltd., Heckmondwike, UK), column heater/ chiller (Jones Chromatography), and DG440 in-line degasser (Phenomenex, UK). Analytical data were collected using Borwin HPLC software (version 1.5), For the detection of DMU215, an in-line photoreactivity cell was used (Beam Boost^TM^, Advanced Separations Technology Incorporated).

## 5. Conclusions

In an attempt to find the structure–activity relationship for stilbenoids as potential anticancer drugs, a library of novel and reference stilbenoids was designed by mapping onto the steroidal framework of oestradiol and known tyrosine kinase inhibitors such as resveratrol. A polymethoxy *trans* stilbene, DMU212, was identified as a potent selective cytotoxic agent, inhibiting the growth of breast tumour cell lines MDA468 and MCF7 cells at sub-micromolar doses, but showing little toxicity towards non-tumour breast cell line MCF10A at these concentrations.

## Data Availability

Data are contained within the article and Appendix A.

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
