# Peer review of "CYP1-Activation and Anticancer Properties of Synthetic Methoxylated Resveratrol Analogues"

_molecules, 2024, doi:10.3390/molecules29020423_

Round 1
Reviewer 1 Report
Comments and Suggestions for Authors
Authors synthesized (E)-stilbenoid resveratrol and (Z)-stilbenoid combretastatin A, assessed their cytotoxic activity by the MTT assay and studied CYP1 mediated bioconversion of resveratrol analog DMU212. The metabolic products of DMU212 were confirmed by comparing synthetic compounds and LCMS co-elution studies. Compounds were characterized by 1H NMR, 13C NMR and HRMS. This introduction, result and discussion sections are clearly written in this manuscript. I would recommend this manuscript may be considered for publication in molecules journal.
Minor comments in characterization data
Did HRMS for compound DMU205 show [M]+? I think it should be [M+H]+. Authors can check for some other compounds in experimental section where it is written as [M]+. There is error greater than 5 ppm for HRMS of compound DMU 271. Authors can reverify these with respective spectra.
Author Response
Reviewer 1
Authors synthesized (E)-stilbenoid resveratrol and (Z)-stilbenoid combretastatin A, assessed their cytotoxic activity by the MTT assay and studied CYP1 mediated bioconversion of resveratrol analog DMU212. The metabolic products of DMU212 were confirmed by comparing synthetic compounds and LCMS co-elution studies. Compounds were characterized by 1H NMR, 13C NMR and HRMS. This introduction, result and discussion sections are clearly written in this manuscript. I would recommend this manuscript may be considered for publication in molecules journal.
Thank you for your encouraging remarks.
Minor comments in characterization data
Did HRMS for compound DMU205 show [M]+? I think it should be [M+H]+. Authors can check for some other compounds in experimental section where it is written as [M]+.
We checked the data and found that they were correct. We had our accurate mass determination done by the Mass Spectrometry Service at Swansea University. They used different ionisation modes, and as a result, sometimes we received the [M]+ data and sometimes the [M+H]+ data. We were only provided with the end results, and now with hindsight it is not feasible to collect the ionisation modes that were used then.
We went through the manuscript, and changed any mention of [M+1]+ by the more correct [M+H]+ (the changes are highlighted in the revised manuscript)
There is error greater than 5 ppm for HRMS of compound DMU 271. Authors can reverify these with respective spectra.
This was a typing error: C17H19O3 requires [M+H]+ 271.1329. The correction is highlighted in the revised version>
Reviewer 2 Report
Comments and Suggestions for Authors
The paper entitled „CYP1-activation and anticancer properties of synthetic methoxylated resveratrol analogues” is outstanding piece of broad designed anticancer (anti BC) activity of cis- and trans- methoxylated stilbenoids.
The Introduction allows potential reader to understand the entire concept of performed studies. The chemistry involved in syntheses is described in details both in main text and well-documented in Supplementary materials.
The conclusion on potential usefulness of DMU 212 as pro-drug, converted into highly toxic DMU 214 and DMU 283 against MCF-7 (TCDD) is promising, especially because high TS evidenced.
I strongly recommend to publish this article. Some corrections should be introduced as follows:
Line 179: is DMU313to; should be rather DMU313 to;
Line 290: please exchange hypothesis instead of theory
Line 365: is: and the reaction was stirred; it is better to say: and the mixture was stirred
Line 389: close the bracket after 11-19Hz
Line 408: by of the ?
Appendix A, 13-C NMR spectrum of (E)-4,4’-dimethoxystilbene (DMU 205): number of 13-C resonances is 5; rather it should be 6
General remark: units like microM concentration or mg should be spaced after numbers given (lines: 155, 374, 389, 469, 518, 355, 358); not necessarily in Appendix.
What an excellent job; I have learned a lot from it. Thank you
Author Response
Reviewer 2
Comments and Suggestions for Authors
The paper entitled „CYP1-activation and anticancer properties of synthetic methoxylated resveratrol analogues” is outstanding piece of broad designed anticancer (anti BC) activity of cis- and trans- methoxylated stilbenoids.
The Introduction allows potential reader to understand the entire concept of performed studies. The chemistry involved in syntheses is described in details both in main text and well-documented in Supplementary materials.
The conclusion on potential usefulness of DMU 212 as pro-drug, converted into highly toxic DMU 214 and DMU 283 against MCF-7 (TCDD) is promising, especially because high TS evidenced.
I strongly recommend to publish this article. Some corrections should be introduced as follows:
Thank you for your supporting remarks. We have addressed the suggested correction, and highlighted them in the revised version of the manuscript.
Line 179: is DMU313to; should be rather DMU313 to;
Corrected, we added the space
Line 290: please exchange hypothesis instead of theory
Corrected as suggested
Line 365: is: and the reaction was stirred; it is better to say: and the mixture was stirred
Corrected as suggested
Line 389: close the bracket after 11-19Hz
Corrected as suggested
Line 408: by of the ?
Changed into: … metabolism of the methylenedioxy substituent…
Appendix A, 13-C NMR spectrum of (E)-4,4’-dimethoxystilbene (DMU 205): number of 13-C resonances is 5; rather it should be 6
Thanks for alerting us to this. We had overlooked one peak resonance (δ 126.00 ppm). This has now been added,
General remark: units like microM concentration or mg should be spaced after numbers given (lines: 155, 374, 389, 469, 518, 355, 358); not necessarily in Appendix.
Corrected as suggested
What an excellent job; I have learned a lot from it. Thank you
Thank you very much.
Reviewer 3 Report
Comments and Suggestions for Authors
This is a well-designed and carefully performed study that emphasizes the role of P450 enzymes in the activation of certain anti-cancer drugs. The most problematic part is the Materials and Methods. It should be better structured, starting with the description of materials and general techniques, followed by general synthetic procedures and biological assays. The description of synthetic procedures appears to miss the first step in Scheme 1 (conversion of benzyl alcohols to alkyl chlorides). Detailed synthetic procedures, presented as an Appendix, should be presented as Supporting Information and placed after the References. The description of MTT assays can be significantly shortened by removing the standard cell culture procedures, such as cell trypsinization and counting. Concentration of the MTT reagent should be specified. Other suggested minor changes are as follows:
1. In Figure 2, the tyrosine kinase used should be specified. The inhibitory concentrations are micromolar, not millimolar. The reference should be given by its number (27).
2. Line 61: insert ‘in’ before ‘brain’.
3. Line 112: insert ‘cell line’ after ‘positive’.
4. Line 159: move the heading of Table 1 to the next page.
5. Line 311: something is wrong with the reference formatting.
Comments on the Quality of English LanguageI've spotted a couple of grammatic errors (see above).
Author Response
Reviewer 3
Comments and Suggestions for Authors
This is a well-designed and carefully performed study that emphasizes the role of P450 enzymes in the activation of certain anti-cancer drugs.
Thank you for the compliment.
The most problematic part is the Materials and Methods. It should be better structured, starting with the description of materials and general techniques, followed by general synthetic procedures and biological assays.
The description of MTT assays can be significantly shortened by removing the standard cell culture procedures, such as cell trypsinization and counting. Concentration of the MTT reagent should be specified.
We went critically through the Materials and Methods section and followed your guidelines. The biological assays are written in a more concise way, and we’ve numbered the sections to better show the coherence,
The description of synthetic procedures appears to miss the first step in Scheme 1 (conversion of benzyl alcohols to alkyl chlorides).
Scheme 1 is a generic scheme, not necessarily covering all the synthetic procedures. Often we could simply buy the alkyl chlorides that were needed. If they were not available (e.g. DMU 212), then an Appel reaction could be used to convert a benzyl alcohol to an alkyl chloride (see section 2.1).
Detailed synthetic procedures, presented as an Appendix, should be presented as Supporting Information and placed after the References.
Thank you. We followed your advice.
Other suggested minor changes are as follows:
- In Figure 2, the tyrosine kinase used should be specified. The inhibitory concentrations are micromolar, not millimolar. The reference should be given by its number (27).
Done as suggested - 2.Line 61: insert ‘in’ before ‘brain’.
Done as suggested - Line 112: insert ‘cell line’ after ‘positive’.
Done as suggested - Line 159: move the heading of Table 1 to the next page.
Done as suggested - Line 311: something is wrong with the reference formatting.
Corrected